# End-to-End Neural Relation Extraction with Global Optimization

## Abstract

Neural networks have recently shown promising results for relation extraction. State-of-the-art models cast the task as an end-to-end problem, solved incrementally using a local classifier. Yet previous work using statistical models have demonstrated that global optimization can achieve better performances compared to local classification. We build a globally optimized neural model for end-to-end relation extraction, proposing novel LSTM features in order to better learn representations. Experiments show that our model is highly effective, achieving the best performances on two standard benchmarks.

## 1 Introduction

Extracting entities (Florian et al., 2006, 2010) and relations (Zhao and Grishman, 2005; Jiang and Zhai, 2007; Sun et al., 2011; Plank and Moschitti, 2013) from unstructured texts have been two central tasks in information extraction (Grishman, 1997; Doddington et al., 2004). Traditional approaches to relation extraction take entity recognition as a predecessor step in a pipeline (Zelenko et al., 2003; Chan and Roth, 2011), predicting relations between given entities.

In recent years, there has been a surge of interest in performing end-to-end relation extraction, jointly recognizing entities and relations given free text inputs (Li and Ji, 2014; Miwa and Sasaki, 2014; Miwa and Bansal, 2016; Gupta et al., 2016). End-to-end learning prevents error propagation in the pipeline approach, and allows cross-task dependencies to be modeled explicitly for entity recognition. As a result, it gives better relation extraction accuracies compared to pipelines.

Miwa and Bansal (2016) were among the first to use neural networks for end-to-end relation extraction, showing highly promising results. In particular, they used a bidirectional LSTM (Graves et al., 2013) to learn hidden word representations under a sentential context, and further leveraged a tree-structured LSTMs (Tai et al., 2015) to encode syntactic information, given the output of a parser. The resulting representations are then used for making local decisions for entity and relation extraction incrementally, leading to much improved results compared with the best statistical model (Li and Ji, 2014). This demonstrates the strength of neural representation learning for end-to-end relation extraction.

On the other hand, Miwa and Bansal (2016)'s model is trained locally, without considering structural correspondences between incremental decisions. This is unlike existing statistical methods, which utilize well-studied structured prediction methods to address the problem (Li and Ji, 2014; Miwa and Sasaki, 2014). As has been commonly understood, learning local decisions for structured prediction can lead to label bias (Lafferty et al., 2001), which prevents globally optimal structures from receiving optimal scores by the model. We address this potential issue by building a structural neural model for end-to-end relation extraction, following a recent line of efforts on globally optimized models for neural structured prediction (Zhou et al., 2015; Watanabe and Sumita, 2015; Andor et al., 2016; Wiseman and Rush, 2016).

In particular, we follow Miwa and Bansal (2016), casting the task as an end-to-end table-filling problem. This is different from the action-based method of Li and Ji (2014), yet has shown to be more flexible and accurate (Miwa and Sasaki, 2014). We take a different approach to representation learning, addressing two potential limitations of Miwa and Bansal (2016).

First, Miwa and Bansal (2016) rely on external syntactic parsers for obtaining syntactic information, which is crucial for relation extraction (Culotta and Sorensen, 2004; Zhou et al., 2005; Bunescu and Mooney, 2005; Qian et al., 2008). However, parsing errors can lead to encoding inaccuracies of tree-LSTMs, thereby hurting relation extraction potentially. We take an alternative approach to integrating syntactic information, by taking the hidden LSTM layers of a bi-affine attention parser (Dozat and Manning, 2016) to augment input representations. Pretrained for parsing, such hidden layers contain rich syntactic information on each word, yet do not explicitly represent parsing decisions, thereby avoiding potential issues caused by incorrect parses.

Our method is also free from a particular syntactic formalism, such as dependency grammar, constituent grammar or CCG, requiring only hidden representations on word that contain syntactic information. In contrast, the method of Miwa and Bansal (2016) must consider tree LSTM formulations that are specific to grammar formalisms, which can be very different (Tai et al., 2015).

Second, Miwa and Bansal (2016) did not explicitly learn the representation of segments when predicting entity boundaries or making relation classification decisions, which can be intuitively highly useful. We take the LSTM-Minus method of Wang and Chang (2016), modelling a segment as the difference between its last and first LSTM hidden vectors. This method is highly efficient, yet gives as accurate results as compared to more complex neural network structures to model a span of words (Cross and Huang, 2016).

Evaluation on two benchmark datasets shows that our method outperforms previous methods of Miwa and Bansal (2016), Li and Ji (2014) and Miwa and Sasaki (2014), giving the best reported results on both benchmarks. Our code is available under GPL at https://github.com/⟨anonymized⟩.

## 2 Model

### 2.1 Task Definition

As shown in Figure 1, the goal of relation extraction is to mine relations from raw texts. It consists of two sub-tasks, namely entity detection, which recognizes valid entities, and relation classification, which determines the relation categories over entity pairs. We follow recent studies and recognize entities and relations as one single task.

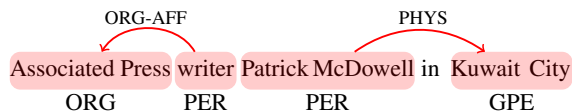

Figure 1: Relation extraction. The example is chosen from the ACE05 dataset, where ORG, PER and GPE denote organization, person and geo-political entities, respectively; ORG-AFF and PHYS denote organization affiliation and physical relations, respectively.

### 2.2 Method

We follow Miwa and Sasaki (2014) and Gupta et al. (2016), treating relation extraction as a table-filling problem, performing entity detection and relation classification using a single incremental model, which is similar in spirit to Miwa and Bansal (2016). Figure 2 shows an example of the table-filling process.

Formally, given a sentence $w_1 w_2 \cdots w_n$, we maintain a table $T^{n \times n}$, where $T(i, j)$ denotes the relation between $w_i$ and $w_j$. When $i = j$, $T(i, j)$ denotes an entity boundary label. We map entity words into labels under the BILOU (Begin, Inside, Last, Outside, Unit) scheme, assuming that there are no overlapping entities in one sentence (Li and Ji, 2014; Miwa and Sasaki, 2014; Miwa and Bansal, 2016). Only the upper triangular table is necessary for indicating the relations.

We adopt the close-first left-to-right strategy (Miwa and Sasaki, 2014) to map the two-dimensional table into a sequence, in order to fill the table incrementally. During the table-filling process, we take two label sets for entity detection ($i = j$) and relation classification ($i < j$), respectively. The labels for entity detection include {B-*, I-*, L-*, O, U-* }, where * denotes the entity type, and the labels for relation classification are $\{\overrightarrow{*}, \overleftarrow{*}\}$, where * denotes the relation category.

At each step, given a partially-filled table $T$, we determine the most suitable label $l$ for the next step (order in Figure 2) using a scoring function:

$$\text{score}(T, l) = W_l h_T, \qquad (1)$$

where $W_l$ is a model parameter and $h_T$ is the vector representation of $T$. Based on the function, we aim to find the best label sequence $l_1 \cdots l_m$, where $m = \frac{n(n+1)}{2}$, and the resulting sequence of partially-filled tables is $T_0 T_1 \cdots T_m$, where $T_i =$ FILL$(T_{i-1}, l_i)$, and $T_0$ is an empty table. Different from previous work, we investigate a structural

|  | Associated | Press | writer | Patrick | McDowell | in | Kuwait | City |
|---|---|---|---|---|---|---|---|---|
| Associated | 1 B-ORG | 9 ⊥ | 16 ⊥ | 22 ⊥ | 27 ⊥ | 31 ⊥ | 34 ⊥ | 36 ⊥ |
| Press |  | 2 L-ORG | 10 $\overleftarrow{\text{ORG-AFF}}$ | 17 ⊥ | 23 ⊥ | 28 ⊥ | 32 ⊥ | 35 ⊥ |
| writer |  |  | 3 U-PER | 11 ⊥ | 18 ⊥ | 24 ⊥ | 29 ⊥ | 33 ⊥ |
| Patrick |  |  |  | 4 B-PER | 12 ⊥ | 19 ⊥ | 25 ⊥ | 30 ⊥ |
| McDowell |  |  |  |  | 5 L-PER | 13 ⊥ | 20 ⊥ | 26 $\overrightarrow{\text{PHYS}}$ |
| in |  |  |  |  |  | 6 O | 14 ⊥ | 21 ⊥ |
| Kuwait |  |  |  |  |  |  | 7 B-GPE | 15 ⊥ |
| City |  |  |  |  |  |  |  | 8 L-GPE |

Figure 2: Table-filling example, where numbers indicate the filling order.

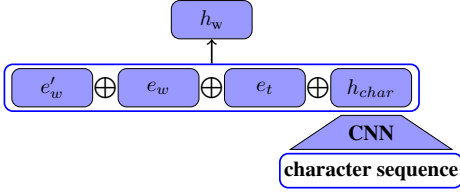

Figure 3: Word representations.

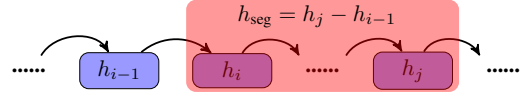

Figure 4: Segment representation.

model that is optimized for the label sequence $l_1 \cdots l_m$ globally, rather than for each $l_i$ locally.

### 2.3 Representation Learning

At the $i$th step, we determine the label $l_i$ of the next table slot based on the current hypothesis $T_{i-1}$. Following Miwa and Bansal (2016), we use a neural network to learn the vector representation of $T_{i-1}$, and then use Equation 1 to rank candidate next labels. There are two types of input features, including the word sequence $w_1 w_2 \cdots w_n$, and the readily filled label sequence $l_1 l_2 \cdots l_{i-1}$. We build a neural network to represent $T_{i-1}$.

#### 2.3.1 Word Representation

Shown in Figure 3, we represent each word $w_i$ by $h_i^w$ using its word form, POS tag and characters. Based on the word form, we use two different embeddings, one being obtained by using a randomly initialized look-up table $E_w$, tuned during training and represented by $e_w$, and the other being a pre-trained external word embedding from $E'_w$, which is fixed and represented by $e'_w$.[1] For a POS tag $t$, its embedding $e_t$ is obtained from a look-up table $E_t$ similar to $E_w$.

The above two components have also been used by Miwa and Bansal (2016). In addition, we enhance the word representation by its character sequence (Ballesteros et al., 2015; Lample et al., 2016), using a convolution neural network (CNN) to derive a character-based word representation $h_{char}$, which has been demonstrated effective for

---

[1] We use the set of pre-trained glove word embeddings available at http://nlp.stanford.edu/data/glove.6B.zip as external word embeddings.

several NLP tasks (dos Santos and Gatti, 2014). We obtain the final $h_i^w$ based on a non-linear feed-forward layer on $e'_w \oplus e_w \oplus e_t \oplus h_{char}$, where $\oplus$ denotes concatenation.

#### 2.3.2 Label Representation

In addition to the word sequence, the history label sequence $l_1 l_2 \cdots l_{i-1}$, especially the labels representing detected entities, is also useful disambiguation. For example, the previous entity boundary label can be helpful to deciding the boundary label of the current word. During relation classification, the types of the entities involved can indicate the relation category between them. We exploit the diagonal label sequence of partial table $T$, which denotes entity boundaries of words, to enhance the representation learning. We obtain a word's entity boundary label embedding $e_l$ by a randomly initialized looking-up table $E_l$.

#### 2.3.3 LSTM Features

We follow Miwa and Bansal (2016), learning global context representations using LSTMs. Three *basic* LSTM structures are used: a left-to-right word LSTM ($\overrightarrow{\text{LSTM}}_w$), a right-to-left word LSTM ($\overleftarrow{\text{LSTM}}_w$) and a left-to-right entity boundary label LSTM ($\overrightarrow{\text{LSTM}}_e$). Each LSTM derives a sequence of hidden vectors for inputs. For example, for $w_1 w_2 \cdots w_n$, $\overrightarrow{\text{LSTM}}_w$ gives $h_1^{w,\rightarrow} h_2^{w,\rightarrow} \cdots h_n^{w,\rightarrow}$.

Different from Miwa and Bansal (2016), who use the output hidden vectors $\{h_i\}$ of LSTMs to represent words, we exploit *segment* representations as well. In particular, for a segment of text $[i, j]$, the representation is computed by using LSTM-Minus (Wang and Chang, 2016), shown by Figure 4, where $h_j - h_{i-1}$ in a left-to-right LSTM

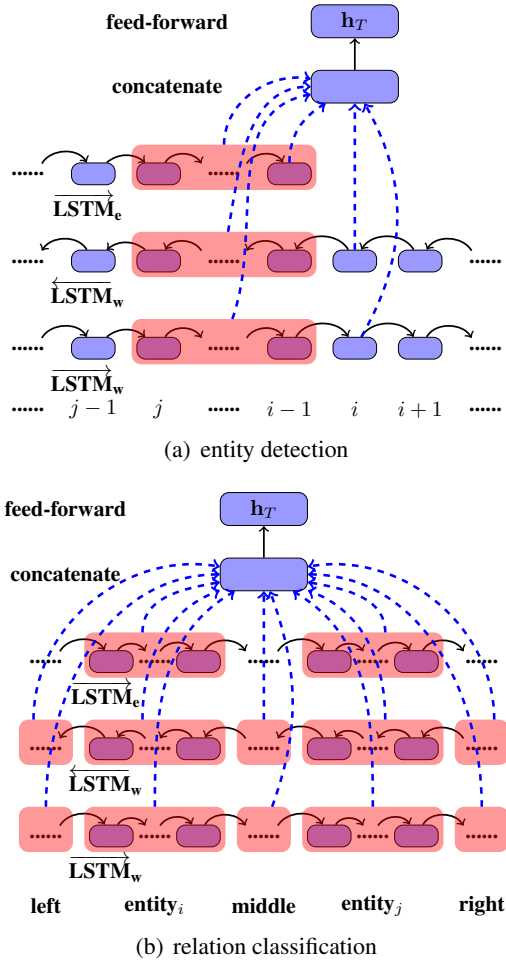

(a) entity detection

(b) relation classification

Figure 5: Feature representation.

and $h_i - h_{j+1}$ in a right-to-left LSTM are used to represent the segment $[i, j]$. The segment representations can reflect entities in a sentence, and thus can be potentially useful for both entity detection and relation extraction.

### 2.3.4 Feature Representation

We use separate feature representations for entity detection and relation classification, both of which are induced according to the above three LSTM structures. In particular, we first extract a set of base neural features, and then concatenate them and feed them into a non-linear neural layer for entity detection and relation classification, respectively. Figure 5 shows the overall representation.

[**Entity Detection**] Figure 5(a) shows the feature representation method for the entity detection actions. First, we extract six feature vectors from the three basic LSTMs, three of which are word features, namely $h_i^{w,\rightarrow}$, $h_i^{w,\leftarrow}$ and $h_{i-1}^{e,\rightarrow}$, and the remaining are segment features, namely $h_{[j,i-1]}^{w,\rightarrow}$, $h_{[j,i-1]}^{w,\leftarrow}$ and $h_{[j,i-1]}^{e,\rightarrow}$, where $j$ denotes the start po-

sition of the previous entity.[2] The six vectors are concatenated and then fed into a non-linear layer for entity detection.

[**Relation Classification**] Figure 5(b) shows the feature representation method for relation classification. Similar to entity detection, we extract a set of features from the basic LSTMs ($\overrightarrow{\text{LSTM}}_w$, $\overleftarrow{\text{LSTM}}_w$ and $\overrightarrow{\text{LSTM}}_e$), and then concatenate them for a non-linear classification layer. The differences between relation classification with entity detection lie in the range of hidden layers from LSTMs. For relation classification between $i$ and $j$, we split each LSTM into five segments according to the two entities ended with $i$ and $j$. Formally, let $[s(i), i]$ and $[s(j), j]$ denote the two entities above, where $s(\cdot)$ denotes the start position of an entity, the resulted segments are $[0, s(i) - 1]$ (i.e., **left**, in Figure 5(b)), $[s(i), i]$ (i.e., **entity**$_i$), $[i + 1, s(j) - 1]$ (i.e., **middle**), $[s(j), j]$ (i.e., **entity**$_j$) and $[j + 1, n]$ (i.e., **right**), respectively. For the word LSTMs, we extract all five segment features, while the entity label LSTM, we only use the segment features of **entity**$_i$ and **entity**$_j$.

### 2.3.5 Syntactic Features

Previous work has shown that syntactic features are useful for relation extraction (Zhou et al., 2005). For example, the shortest dependency path has been used by several relation extraction models (Bunescu and Mooney, 2005; Miwa and Bansal, 2016). Here we propose a novel method to integrate syntax, without need for prior knowledge on concrete syntactic structures.

In particular, many state-of-the-art syntactic parsers use encoder-decoder neural models (Buys and Blunsom, 2015; Kiperwasser and Goldberg, 2016), where the encoder represents the features of the input sentences. For example, LSTM over the input word/tag sequences has been used frequently (Kiperwasser and Goldberg, 2016). The decoder can also leverage partially-parsed results, such as features from partial syntactic trees. Table 1 shows the encoder structures of three state-of-the-art dependency parsers.

Our method is to dump the encoder source representations of state-of-the-art parsers, and then use them directly as part of input embeddings in our proposed model. Denoting the dumped syntactic features on each word as $h_1^{\text{syn}} h_2^{\text{syn}} \cdots h_n^{\text{syn}}$, we feed them into a non-linear neural layer,

---

[2]The non-entity word is treated as a special unit entity to extract segmental features.

| Models | Encoder | LAS |
|--------|---------|-----|
| S-LSTM (2015) | 1-Layer LSTM | 90.9 |
| K&G (2016) | 2-Layer Bi-LSTM | 91.9 |
| D&M (2016) | 4-Layer Bi-LSTM | **93.8** |

Table 1: The encoder structures and performances of three state-of-the-art dependency parsers, where S-LSTM (2015) refers to Dyer et al. (2015), K&G (2016) refers to the best parser of Kiperwasser and Goldberg (2016), D&M (2016) refers to Dozat and Manning (2016), and LAS (labeled attachment score) is the major evaluation metric for dependency parsing.

and then generate two LSTMs (bi-directional) based on the outputs, namely $\overrightarrow{\text{LSTM}}_{syn}$ and $\overleftarrow{\text{LSTM}}_{syn}$, respectively, augmenting the original three LSTMs into five LSTMs. Features are extracted from the two new LSTMs in the same way as from the basic bi-directional word LSTMs.

We exploit the parser of Dozat and Manning (2016) to extract syntactic features, since it achieves the current best performance for dependency parsing. Our method can be easily generalized to the use of other parsers, which are potentially useful for our task as well. For example, we can use a constituent parser in the same way.

## 2.4 Training and Search

### 2.4.1 Local Optimization

Previous work (Miwa and Bansal, 2016; Gupta et al., 2016) trains model parameters by modeling each step for labeling one input sentence separately. Given a partial table $T$, we first obtain its neural representation $h_T$, and then compute the next label scores $\{l_1, l_2, \cdots, l_s\}$ using Equation 1. The output scores are regularized into a probability distribution $\{p_{l_1}, p_{l_2}, \cdots, p_{l_s}\}$ by using a softmax layer. Our training objective is to minimize the cross-entropy loss between this output distribution with the gold-standard distribution:

$$\text{loss}(T, l_i^g, \Theta) = -\log p_{l_i^g}, \qquad (2)$$

where $l_i^g$ is the gold-standard next label for $T$, and $\Theta$ is the set of all model parameters. We refer this training method as *local optimization*, because it maximizes the score of the gold-standard label at each step locally.

During the decoding phase, the greedy search strategy is applied in consistence with the training. At each step, we find the highest-scored label

---

**Algorithm 1** Beam-search.

$agenda \leftarrow \{ (\textit{empty table}, \text{score}=0.0) \}$
**for** $i$ **in** $1 \cdots$ max-step
$\quad next\_scored\_tables \leftarrow \{ \}$
$\quad$ **for** $scored\_table$ **in** $agenda$
$\quad\quad labels \leftarrow$ NEXTLABELS($scored\_table$)
$\quad\quad$ **for** next_label **in** labels
$\quad\quad\quad$ new $\leftarrow$ FILL(scored_table, $next\_label$)
$\quad\quad\quad$ ADDITEM($next\_scored\_tables$, new)
$\quad agenda \leftarrow$ TOP-B($next\_scored\_tables$, $B$)

---

based on the current partial table, before going on to the next step.

### 2.4.2 Global Optimization

We exploit the global optimization strategy of Andor et al. (2016), maximizing the cumulative score of the gold-standard label sequence for one sentence as a unit. Global optimization has achieved success for several NLP tasks under the neural setting (Zhou et al., 2015; Watanabe and Sumita, 2015). For relation extraction, global learning gives the best performances under the discrete setting (Li and Ji, 2014; Miwa and Sasaki, 2014). We study such models here under the neural setting.

Given a label sequence of $l_1 l_2 \cdots l_i$, the score of $T_i$ is defined as follows:

$$\begin{aligned} \text{score}(T_i) &= \sum_{j=0}^{i} \text{score}(T_{j-1}, l_j) \\ &= \text{score}(T_{i-1}) + \text{score}(T_{i-1}, l_i), \end{aligned} \qquad (3)$$

where $\text{score}(T_0) = 0$ and $\text{score}(T_{i-1}, l_i)$ is computed by Equation 1. By this definition, we maximize the scores of all gold-standard partial tables.

Again cross-entropy loss is used to perform model updates. At each step $i$, the objective function is defined by:

$$\begin{aligned} \text{loss}(x, T_i^g, \Theta) &= -\log p_{T_i^g} \\ &= -\log \frac{\text{score}(T_i^g)}{\sum_{T_i'} \text{score}(T_i')}, \end{aligned} \qquad (4)$$

where $x$ denotes the input sentence, $T_i^g$ denotes the gold-standard state at step $i$, and $T_i'$ are all partial tables that can be reached at step $i$.

The major challenge is to compute $p_{T_i^g}$, because we cannot traverse all partial tables that are valid at step $i$, since their count increases exponentially by the step number. We follow Andor

et al. (2016), approximating the probability by using beam search and early-update.

Shown in Algorithm 1, we use standard beam search, using an agenda to maintain $B$ highest-scored partially-filled tables at each step. When each action of table filling is taken, all hypotheses in the agenda are expanded by enumerating the next labels, and the $B$ highest-scored resulting tables are used to replace the agenda for the next step. Search begins with the agenda containing an empty table, and finishes when all cells of the tables in the agenda have been filled. When the beam size is 1, the algorithm is the same as greedy decoding. When the beam size is larger than 1, however, error propagation from greedy steps can be alleviated. For training, the same beam search algorithm is applied to training examples, and early-update (Collins and Roark, 2004) is used to fix search errors.

## 3 Experiments

### 3.1 Data and Evaluation

We evaluate the proposed model on two datasets, namely the ACE05 data and the corpus of Roth and Yih (2004) (CONLL04), respectively. The ACE05 data defines seven coarse-grained entity types and six coarse-grained relation categories, while the CONLL04 data defines four entity types and five relation categories.

For the ACE05 dataset, we follow Li and Ji (2014) and Miwa and Bansal (2016), splitting and preprocessing the dataset into training, development and test sets.[3] For CONLL04, we follow Miwa and Sasaki (2014) to split and preprocess the dataset into training and test corpus, and divide 10% of the training corpus for development.

We use the micro F1-measure as the major metric to evaluate model performances, treating an entity as correct when its head region and type are both exactly matched,[4] and regard a relation as correct when the argument entities and the relation category are all correct.

### 3.2 Parameter Tuning

We update all model parameters by back propagation using Adam (Kingma and Ba, 2014) with a learning rate $10^{-3}$, using gradient clipping by

---

[3]https://github.com/tticoin/LSTM-ER/.
[4]For the ACE05 dataset, the head region is defined by the corpus, and for the CONLL04 dataset, the head region covers the entire scope of an entity.

| Network Structure | Size |
|---|---|
| Word Embedding | 200 |
| Tag Embedding | 50 |
| Char Embedding | 50 |
| Entity Label Embedding | 50 |
| Input/Output of Word LSTMs | 250 |
| Input/Output of Entity Label LSTMs | 100 |
| Table Representation | 300 |

Table 2: Dimension sizes.

| Model | Entity | Relation |
|---|---|---|
| baseline | **81.5** | **50.9** |
| -character | 80.9 | 50.2 |
| -segment (entity detection) | 80.2 | 49.8 |

Table 3: Feature ablation tests.

a max norm 10 and $l_2$-regularization by a parameter $10^{-5}$. The dimension sizes of various vectors in neural network structure are shown in Table 2. All the hyper-parameters are tuned by development experiments. All experiments are conducted under gcc version 4.9.4 (Ubuntu 4.9.4-2ubuntu1 14.04.1), on an Intel(R) Xeon(R) CPU E5-2670 @ 2.60GHz.

Online training is used to learn model parameters, traversing over the entire training examples by 300 iterations. We select the best iteration model according to the development results. In particular, we exploit pre-training techniques to learn better model parameters (Wiseman and Rush, 2016). For the local model, we follow Miwa and Bansal (2016), training parameters only for entity detection during the first 20 iterations. For the global model, we pretrain our model using local optimization for 40 iterations, before conducting beam global optimization.

### 3.3 Development Experiments

We conduct several development experiments on the ACE05 development dataset.

#### 3.3.1 Feature Ablation Tests

We consider the baseline system with no syntactic features using local training. Compared with Miwa and Bansal (2016), we introduce character-level features, and in addition exploit segmental features for entity detection. Feature ablation experiments are conducted for the two types of features. As shown in Table 3, without character-level features, the F-scores of entity detection and relation classification decrease 0.6% and 0.7%, respectively. Without segmental features for entity

| Model | Beam | F1 | Speed |
|---|---|---|---|
| Local | 1 | 50.9 | **95.6** |
| Local(+SS) | 1 | 51.2 | 95.1 |
| Global | 1 | 51.4 | 95.3 |
| | 3 | 51.8 | 52.0 |
| | 5 | **52.6** | 36.9 |

Table 4: Comparisons between local and global models, where SS denotes scheduled sampling, and speed is measured by the number of sentences per second.

detection, the baseline model loses 1.3% in entity detection, which results in an error propagation of 1.1% for relation classification. The results demonstrate that the two types of new features we use are useful for relation extraction.

### 3.3.2 Local v.s. Global

We study the influence of training strategies for the relation extraction model without using syntactic features. For the local model, we apply the scheduled sampling strategy (Bengio et al., 2015), which has been shown to improve the performances by Miwa and Bansal (2016).

Table 4 shows the relation F1 scores. Scheduled sampling achieves improved F-measure scores for the local model. With the same greedy search strategy, the globally normalized model gives slightly better results than the local model with scheduled sampling. The performance of the global model increases with a larger beam size. However, the decoding speed becomes intolerably slow when the beam size increases beyond 5. Thus we exploit a beam size of 5 for global training considering both performance and efficiency.

### 3.3.3 Syntactic Features

We examine the effectiveness of the proposed syntactic features. Table 5 shows the developmental results using both local and global optimization. The proposed features improve the relation performances significantly under both settings, where the p-values are below $10^{-4}$ by using pairwise t-test, demonstrating that our use of syntactic features is highly effective for relation extraction.

### 3.4 Final Results

Table 6 shows the final results on the test datasets of ACE05 and CONLL04. We show several top-performing systems in the table as well, where M&B (2016) refers to Miwa and Bansal (2016), who exploit end-to-end LSTM neural networks

| Model | Features | Entity | Relation |
|---|---|---|---|
| Local | all | **81.6** | **53.0** |
| | -syn | 81.5 | 50.9 |
| Global | all | **81.9** | **54.2** |
| | -syn | 81.6 | 52.6 |

Table 5: The influence of syntactic features.

| model | ACE05 | | CONLL04 | |
|---|---|---|---|---|
| | Entity | Relation | Entity | Relation |
| Our Model | **83.6** | **57.5** | **85.6** | **67.8** |
| M&B (2016) | 83.4 | 55.6 | — | — |
| L&J (2014) | 80.8 | 49.5 | — | — |
| M&S (2014) | — | — | 80.7 | 61.0 |

Table 6: Final results on the test datasets.

with local optimization, and L&J (2014) and M&S (2014) refer to Li and Ji (2014) and Miwa and Sasaki (2014), respectively, which are both globally optimized models using discrete features, giving the top F-scores among statistical models.[5]

Overall, neural models give better performances than statistical models, and global optimization can give improved performances as well. Our final model achieves the best performances on both datasets. Compared with the best reported results, our model gives improvements of 1.9% on ACE05, and 6.8% on CONLL04.

### 3.5 Analysis

We conduct analysis on the ACE05 test dataset in order to understand our models in depth. We focus on two major contributions by our model, first examining the influences of global optimization, and then studying the gains by using the proposed syntactic features.

Global optimization aims to find the best label sequences, rather than the best label locally at each step. Thus intuitively global optimization should give better accuracies at the sentence level. We verify this by examining the sentence-level accuracies, where one sentence is regarded as correct when all the labels in the resulted table are correct. Figure 6 shows the result, which is consistent with our intuition. On the one hand, the sentence-

---

[5]Gupta et al. (2016) proposed a locally optimized model but used a different test dataset from CONLL04 and a different evaluation method, reporting entity and relation F-scores of 93.6% and 72.1%, respectively. Their results are not directly comparable to the results in Table 6. In particular, they regard an entity as correct if at least one token is tagged correctly, which influences the results significantly since multi-word entities accounts for over 50% of all entities.

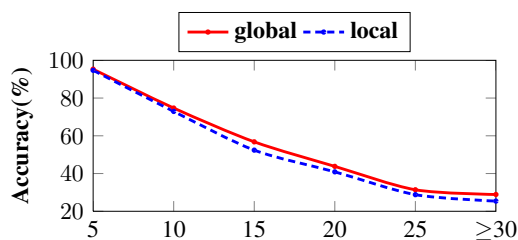

Figure 6: Sentence-level accuracies with respect to sentence length.

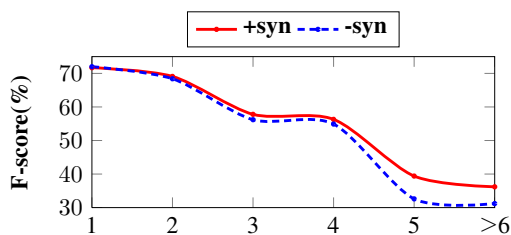

Figure 7: F-scores with respect to the distance between entity pairs.

level accuracies of the globally normalized model are consistently better than the local model. On the other hand, the accuracy decreases sharply as the sentence length increases, with the local model suffering more severely from larger sentences.

To understand the effectiveness of the proposed syntactic features, we examine the relation F-scores with respect to entity distances. Miwa and Bansal (2016) exploit the shortest dependency path, which can make the distance between two entities closer compared with their sequential distance, thus facilitating relation extraction. We verify whether the proposed syntactic features can benefit our model similarly. As shown in Figure 7, the F-scores of entity-pairs with large distances see apparent improvements, demonstrating that our use of syntactic features has a similar effect compared to the shortest dependency path.

## 4 Related Work

Entity recognition (Florian et al., 2004, 2006; Ratinov and Roth, 2009; Florian et al., 2010; Kuru et al., 2016) and relation extraction (Zhao and Grishman, 2005; Jiang and Zhai, 2007; Zhou et al., 2007; Qian and Zhou, 2010; Chan and Roth, 2010; Sun et al., 2011; Plank and Moschitti, 2013; Verga et al., 2016) have received much attention in the NLP community. The dominant methods treat the two tasks separately, where relation extraction is performed assuming that entity boundaries have been given (Zelenko et al., 2003; Miwa et al.,

2009; Chan and Roth, 2011; Lin et al., 2016).

Several studies find that extracting entities and relations jointly can benefit both tasks. Early work conducts joint inference for separate models (Ji and Grishman, 2005; Roth and Yih, 2004, 2007). Recent work shows that joint learning and decoding with a single model brings more benefits for the two tasks (Li and Ji, 2014; Miwa and Sasaki, 2014; Miwa and Bansal, 2016; Gupta et al., 2016), and we follow this line of work in the study.

LSTM features have been extensively exploited for NLP tasks, including tagging (Huang et al., 2015; Lample et al., 2016), parsing (Kiperwasser and Goldberg, 2016; Dozat and Manning, 2016), relation classification (Xu et al., 2015; Vu et al., 2016; Miwa and Bansal, 2016) and sentiment analysis (Li et al., 2015; Teng et al., 2016). Based on the output of LSTM structures, Wang and Chang (2016) introduce segmental features, and apply it to dependency parsing. The same method is applied to constituent parsing by Cross and Huang (2016). We exploit this segmental representation for relation extraction.

Global optimization and normalization has been successfully applied on many NLP tasks that involve structural prediction (Lafferty et al., 2001; Collins, 2002; McDonald et al., 2010; Zhang and Clark, 2011), using traditional discrete features. For neural models, it has recently received increasing interests (Zhou et al., 2015; Andor et al., 2016; Xu, 2016; Wiseman and Rush, 2016), and improved performances can be achieved with global optimization accompanied by beam search. Our work is in line with these efforts. To our knowledge, we are the first to apply globally optimized neural models for end-to-end relation extraction, achieving the best results on standard benchmarks.

## 5 Conclusion

We proposed a novel relation extraction model using neural network, based on the table-filling framework proposed by Miwa and Sasaki (2014). Feature representations are learned from several LSTM structures over the inputs, and a simple method is used to integrate syntactic information into our model without the need of parser outputs. In addition, global optimization is taken to make use of structural information more effectively. Compared with previous work, our final model achieved the best performances on two benchmark datasets.

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
