# Peer review of "End-to-End Neural Relation Extraction with Global Optimization"

_ACL 2017 — decision unknown_

[Official Review · Reviewer 1 · rating 4 · confidence 5]
soundness 3 · originality 3 · clarity 5 · impact 4 · substance 4 · appropriateness 5 · meaningful comparison 4 · presentation format Poster

- Strengths:

 - The paper is clearly written and well-structured. 

 - The system newly applied several techniques including global optimization to
end-to-end neural relation extraction, and the direct incorporation of the
parser representation is interesting.

 - The proposed system has achieved the state-of-the-art performance on both
ACE05 and CONLL04 data sets.

 - The authors include several analyses.

- Weaknesses:

 - The approach is incremental and seems like just a combination of existing
methods.  

 - The improvements on the performance (1.2 percent points on dev) are
relatively small, and no significance test results are provided.

- General Discussion:

- Major comments:

 - The model employed a recent parser and glove word embeddings. How did they
affect the relation extraction performance?

 - In prediction, how did the authors deal with illegal predictions?

- Minor comments:

 - Local optimization is not completely "local". It "considers structural
correspondences between incremental decisions," so this explanation in the
introduction is misleading.

 - Points in Figures 6 and 7 should be connected with straight lines, not
curves.

 - How are entities represented in "-segment"?

 - Some citations are incomplete. Kingma et al. (2014) is accepted to ICLR,
and Li et al. (2014) misses pages.